# UNIFIED NEURAL REPRESENTATION MODEL FOR PHYSICAL SPACE AND LINGUISTIC CONCEPTS

## ABSTRACT

The spatial processing system of the brain uses grid-like neural representations (grid cells) for supporting vector-based navigation. Experiments also suggest that neural representations for concepts (concept cells) exist in the human brain, and conceptual inference relies on navigation in conceptual spaces. We propose a unified model called "disentangled successor information (DSI)" that explains neural representations for physical space and linguistic concepts. DSI generates grid-like representations in a 2-dimensional space that highly resemble those observed in the brain. Moreover, the same model creates concept-specific representations from linguistic inputs, corresponding to concept cells. Mathematically, DSI vectors approximate value functions for navigation and word vectors obtained by word embedding methods, thus enabling both spatial navigation and conceptual inference based on vector-based calculation. Our results suggest that representations for space and concepts can emerge from a shared mechanism in the human brain.

## 1 INTRODUCTION

In the brain, grid cells in the entorhinal cortex (EC) represent the space by grid-like representations (Hafting et al., 2005; Doeller et al., 2010; Jacobs et al., 2013). This neural representation is often related to vector-based spatial navigation because grid cells provide global metric over the space. Theoretically, an animal can estimate the direction to a goal when representations of a current position and a goal position are given (Fiete et al., 2008; Bush et al., 2015). Furthermore, self-position can be estimated by integrating self-motions when sensory information is not available (McNaughton et al., 2006). These functions are the basis of robust spatial navigation by animals.

There are not only spatial but also conceptual representations in EC. Neurons called as "concept cells" have been found in human medial temporal lobe including EC (Quiroga, 2012; Reber et al., 2019). Concept cells respond to specific concepts, namely, stimuli related to a specific person, a famous place, or a specific category like "foods" and "clothes". Furthermore, recent experiments also suggest that grid-like representations appear not only for physical space but also for conceptual space if there is a 2-dimensional structure (e.g. lengths of a neck and legs, intensity of two odors), and those representations are the basis of vector-based conceptual inference (Bao et al., 2019; Constantinescu et al., 2016; Park et al., 2021). Thus, it is expected that there is a shared processing mechanism for physical and conceptual spaces in EC. Existence of shared neural mechanism may also explain why humans use sense of physical space (such as directionality) to communicate abstract concepts (conceptual metaphor (Lakoff & Johnson, 1980)). However, a principle behind such universal computation in the brain is still unclear.

In this paper, we propose a representation model which we call disentangled successor information (DSI) model. DSI is an extension of successor representation (SR), which stems from a theory of reinforcement learning and became one of promising computational models of the hippocampus and EC (Dayan, 1993; Stachenfeld et al., 2017; Momennejad et al., 2017; Momennejad, 2020). Like eigenvectors of SR, DSI forms grid-like codes in a 2-D space, and those representations support vector-based spatial navigation because DSI approximates value functions for navigation in the framework of linear reinforcement learning (Todorov, 2006; 2009; Piray & Daw, 2021). Remarkably, when we apply DSI to text data by regarding a sequence of words as a sequence of states, DSI forms concept-specific representations like concept cells. Furthermore, we show mathematical correspondence between DSI and word embedding models in natural language processing (NLP)

(Mikolov et al., 2013a;b; Pennington et al., 2014; Levy & Goldberg, 2014), thus we can perform intuitive vector-based conceptual inference as in those models. Our model reveals a new theoretical relationship between spatial and linguistic representation learning, and suggests a hypothesis that there is a shared computational principle behind grid-like and concept-specific representations in the hippocampal system.

## 2 Contributions and related works

We summarize contributions of this work as follows. (1) We extended SR to successor information (SI), by which we theoretically connected reinforcement learning and word embedding, thus spatial navigation and conceptual inference. (2) We found that dimension reduction with constraints for grid-like representations (decorrelative NMF) generates disentangled word vectors with concept-specific units, which has not been found previously. (3) Combining these results, we demonstrated that a computational model for grid cells can be extended to represent and compute linguistic concepts in an intuitive and biologically plausible manner, which has not been shown in previous studies.

Our model is an extension of successor representation (SR), which is recently viewed as a plausible model of hippocampus and EC (Dayan, 1993; Stachenfeld et al., 2017; Momennejad et al., 2017; Momennejad, 2020). Furthermore, default representation (DR), which is based on linear reinforcement learning theory, has been also proposed as a model of EC (Piray & Daw, 2021). We show that our model can extract linguistic concepts, which has not been shown for SR and DR. Furthermore, we demonstrate vector-based compositionality of words in our model, which expands the range of compositionality of EC representations (Piray & Daw, 2021) to semantic processing.

Our model produces biologically plausible grid-like representations in 2-D space, which supports spatial navigation. Previous studies have revealed that non-negative and orthogonal constraints are important to obtain realistic grid-like representations (Dordek et al., 2016; Sorscher et al., 2019). Furthermore, recurrent neural networks form grid-like representations through learning path integration, and those representations support efficient spatial navigation (Banino et al., 2018; Cueva & Wei, 2018; Gao et al., 2019). Some of those models have reproduced experimentally observed scaling ratios between grid cell modules (Banino et al., 2018; Sorscher et al., 2019). However, previous models have not been applied to learning of linguistic concepts, or other complex conceptual spaces in real-world data. Whittington et al. (2020) proposed a unified model for spatial and non-spatial cognition. However, their model was applied only to simple graph structures and conceptual specificity like our model was not observed.

Analogical inference by our model is a same function as word embedding methods in NLP (Mikolov et al., 2013a;b; Pennington et al., 2014; Levy & Goldberg, 2014). However, a unique feature of DSI representations is that each dimension of vectors corresponds to a specific concept like concept cells in the human brain (Quiroga, 2012; Reber et al., 2019). Our model provides biological plausible interpretation of word embedding: each word is represented by combination of disentangled conceptual units, inference is recombination of those concepts, and such representations emerge through the same constraints with grid cells. It was recently shown that transformer-based models (Vaswani et al., 2017; Brown et al., 2020), which are currently state-of-the-art models in NLP, generate grid-like representations when applied to spatial learning (Whittington et al., 2022). Similarly to our model, this finding implies the relationship between spatial and linguistic processing in the brain. However, concept-specific representations has not been found in such model. Furthermore, clear theoretical interpretation in this study depends on the analytical solution for skip-gram (Levy & Goldberg, 2014). Such analytical solution is currently unknown for transformer-based models.

## 3 Model

### 3.1 Disentangled successor information

Let us assume $N_s$ discrete states exist in the environment. Successor representation (SR) between two states $s$ and $s'$ is defined as

$$SR(s, s') = E\left[\sum_{t=0}^{\infty} \gamma^t \delta(s_t, s') | s_0 = s\right] = \sum_{t=0}^{\infty} \gamma^t P(s_t = s' | s_0 = s), \tag{1}$$

where $\delta(i,j)$ is Kronecker's delta and $\gamma$ is a discount factor. We describe how we calculate SR in this study in Appendix A.1. SR and its dimension-reduced representations have been viewed as models of hippocampus and entorhinal cortex, respectively (Stachenfeld et al., 2017).

Based on SR, we define successor information (SI) and positive successor information (PSI) as

$$SI(s,s') = \log(SR(s,s')) - \log(P(s')), \tag{2}$$

$$PSI(s,s') = \max\{SI(s,s'), 0\}. \tag{3}$$

In this study, we regard this quantity as a hippocampal model instead of SR (Figure 1A).

Next, we introduce a novel dimension reduction method which we call decorrelative non-negative matrix factorization (decorrelative NMF). Decorrelative NMF can be regarded as a variant of NMF (Lee & Seung, 1999) with additional constraints of decorrelation. By applying decorrelatve NMF to PSI, we obtain representation vectors called as disentangled successor information (DSI), which we regard as a model of EC (Figure 1A). In decorrelative NMF, we obtain $D$-dimensional vectors $\boldsymbol{x}(s)$ and $\boldsymbol{w}(s)$ ($D < N_s$) by minimization of the following objective function

$$J = \frac{1}{2} \sum_{s,s'} \rho(s,s')(PSI(s,s') - \boldsymbol{x}(s) \cdot \boldsymbol{w}(s'))^2$$
$$+ \frac{1}{2}\beta_{cor} \sum_{i \neq j}(Corr(i,j))^2 + \frac{1}{2}\beta_{reg} \sum_{s}(||\boldsymbol{x}(s)||^2 + ||\boldsymbol{w}(s)||^2), \tag{4}$$

subject to non-negative constraints $\forall i, x_i(s) \geq 0, w_i(s) \geq 0$. $\rho(s,s')$ is a weight for the square error

$$\rho(s,s') = \frac{1}{N_s V}\left(\frac{1}{M}PSI(s,s') + \rho_{min}\right), \tag{5}$$

where $M$ and $V$ are mean and variance of PSI, respectively, and $\rho_{min}$ is a small value to avoid zero-weight. $Corr(i,j)$ is a correlation between two dimensions in $\boldsymbol{x}(s)$

$$Corr(i,j) = \frac{\sum_s \tilde{x}_i(s)\tilde{x}_j(s)}{\sqrt{\sum_s(\tilde{x}_i(s))^2 \sum_s(\tilde{x}_j(s))^2}}, \tag{6}$$

where $\tilde{x}_i(s) = x_i(s) - \frac{1}{N_s}\sum_s x_i(s)$. The first term of the objective function is weighted approximation error minimization, the second term works for decorrelation between dimensions, and the third term regularizes representation vectors. Optimization was performed by Nesterov's accelerated gradient descent method (Nesterov, 1983) with rectification of $x_i(s), w_i(s)$ every iteration. We describe additional details in Appendix A.2.

## 3.2 RELATIONSHIPS WITH REINFORCEMENT LEARNING AND WORD EMBEDDING

We show dual interpretation of our model. On the one hand, DSI approximates value estimation of linear reinforcement learning, thus support goal-directed decision making and navigation. On the

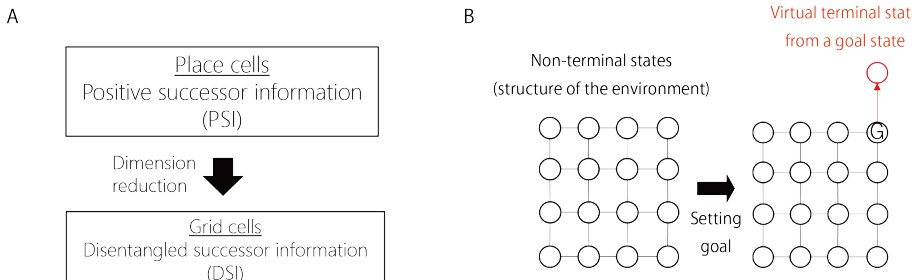

Figure 1: Interpretation of our model. (A) Biological interpretation of our model. PSI as a model of the hippocampus and DSI as a model of EC. (B) The setting of a state transition map in which DSI approximates value functions for spatial navigation.

other hand, the same representation approximates word embedding in NLP, thus support semantic computation.

First, our model approximates value functions of linear reinforcement learning (Todorov, 2006; 2009; Piray & Daw, 2021) in the setting of spatial navigation. Linear reinforcement learning assumes default policy and imposes additional penalty on deviation from default policy, then we can obtain value functions explicitly by solving linear equations. Let us consider a specific condition in which the environment consists of non-terminal states, and a virtual terminal state is attached to a goal state $s_G$ arbitrarily chosen from non-terminal states (Figure 1B). When the agent gets to the goal, it transits to the terminal state with a probability $p_{NT}$. Furthermore, we assume that reward at non-terminal states are uniformly negative and reward at the terminal state is positive so that the agent has to take a short path to goal to maximize reward. In this setting, we can obtain value functions $v^*(s)$ in linear reinforcement learning as

$$\lambda^{-1}v^*(s) = \log(SR^d(s, s_G)) - \log P^d(s_G) = SI^d(s, s_G) \approx \boldsymbol{x}(s) \cdot \boldsymbol{w}(s_G), \tag{7}$$

where $SR^d(s, s_G)$ and $SI^d(s, s_G)$ are SR and SI under the default policy, respectively. We describe details of derivation in Appendix A.3. Therefore, SI is proportional to value functions for spatial navigation and inner products of DSI vectors approximates value functions. Based on this interpretation, we basically regard $\boldsymbol{x}(s)$ as a representation of each state, and $\boldsymbol{w}(s)$ represents a temporary goal.

Second, DSI is related to word embedding methods in NLP (Mikolov et al., 2013a;b; Pennington et al., 2014; Levy & Goldberg, 2014). In linguistics, pointwise mutual information (PMI) and positive pointwise mutual information (PPMI) are used to measure the degree of coincidence between two words (Levy & Goldberg, 2014). They are defined as

$$PMI = \log\left(\frac{P(word_i, word_j)}{P(word_i)P(word_j)}\right), \tag{8}$$

$$PPMI = \max\{PMI, 0\}, \tag{9}$$

where $P(word_i, word_j)$ is a coincidence probability of two words (in a certain temporal window). It has been proven that dimension reduction of PMI approximates a word embedding method skip-gram (Mikolov et al., 2013a;b), and similar performance is obtained using PPMI (Levy & Goldberg, 2014). GloVe (Pennington et al., 2014) is also based on this perspective. SI can be written as

$$SI(s, s') = \log(SR(s, s')) - \log(P(s')) = \log\left(\frac{\sum_{t=0}^{\infty} \gamma^t P(s_t = s', s_0 = s)}{P(s)P(s')}\right). \tag{10}$$

In this formulation, we can see mathematical similarity between PMI and SI by regarding words as states ($s = word_i, s' = word_j$), thus the correspondence between PPMI and PSI. Because of this relationship, we can expect that DSI, which is obtained through dimension reduction of PSI, has similar properties to word embedding methods. The difference is how to count coincidence: the coincidence in SI is evaluated with an asymmetric exponential kernel as in SR, in contrast that a symmetric rectangular temporal window is often used in typical word embedding (see Appendix A.4 for further detail).

## 3.3 DECORRELATIVE NMF RELATES TO GRID CELLS AND DISENTANGLEMENT

Constraints in decorrelative NMF (non-negativity, decorrelation (or orthogonality), and regularization) are important for generation of grid cells, as shown in previous theoretical studies on grid cells. (Dordek et al., 2016; Cueva & Wei, 2018; Banino et al., 2018; Gao et al., 2019; Sorscher et al., 2019). They are also biologically plausible because neural activity is basically non-negative and decorrelation is possible through lateral inhibition. On the other hand, non-negativity (Oja & Plumbley, 2004) and decorrelation (Hyvärinen & Oja, 2000) are also important for extraction of independent components, and it is known that imposing independence in latent space of deep generative models results in the emergence of disentangled representations for visual features (Higgins et al., 2017). Therefore, in word embedding, we expected that those constraints help emergence of independent and disentangled units for linguistic concepts. Constraints in decorrelative NMF are actually crucial for results obtained in this study (Appendix A.5).

As disentangled visual representations explain single-cell activities in higher-order visual cortex (Higgins et al., 2021), we may similarly interpret conceptual representations in our model as concept cells in the human medial temporal lobe (Quiroga, 2012). Previous studies suggest that each concept cell respond to a specific concept, whereas population-level activity patterns represent abstract semantic structures (Reber et al., 2019). Such property is consistent with the factorized and distributed nature of disentangled representation vectors.

## 4 LEARNING REPRESENTATIONS OF PHYSICAL SPACES

In this section, we empirically show that DSI model forms biologically plausible grid-like representations in a 2-D physical space, and they support spatial navigation. These results also apply to conceptual spaces with the 2-D structure, depending on the definition of states.

### 4.1 LEARNING PROCEDURE

As an environment, we assumed a square room tiled with $30 \times 30$ discrete states (Figure 2A). In each simulation trial, an agent starts at one of those 900 states and transits to one of eight surrounding states each time except that transitions are limited at states along the boundary (the structure was not a torus). Transitions to all directions occur with an equal probability. We performed 500 simulation trials and obtained a sequence of 100,000 time steps in each trial. We calculated occurrence probabilities ($P(s)$) and a successor representation matrix ($SR(s, s')$) of 900 states from those sequences, and calculated PSI and DSI (100-dimensional) as described in the Model section. The discount factor $\gamma$ was set to 0.99. We additionally tested spatial navigation in a structure with separated and interconnected rooms (see Figure 3C). In that case, we used the discount factor $\gamma = 0.999$.

### 4.2 EMERGENCE OF GRID-LIKE REPRESENTATIONS

Here we call each dimension of DSI representation vectors $\boldsymbol{x}(s)$ as a neural "unit", and we regard a value in each dimension at each state as a neural activity (or a neural representation). As shown in Figure 2B, many units exhibited grid-like activity patterns in the space. We performed a gridness analysis that has been used in animal experiments (Sargolini et al., 2006) and found that 51% of units were classified as grid cells. Similarly, 53% of units in $\boldsymbol{w}(s)$ were classified as grid cells.

Furthermore, we checked whether DSI representations in the physical space reproduce a property of biological grid cells. Actual grid cells in the rat brain exhibit multiple discrete spatial scales and the ratio between grid scales of adjacent modules is $\sqrt{2}$ (Stensola et al., 2012). We constructed a distribution of grid scales of DSI units by kernel density estimation, which revealed that multiple peaks of grid scales existed and the ratio between grid scales of adjacent peaks was $\sqrt{2}$ (Figure 2C). These results show that DSI model constructs biologically plausible grid-like representations in the 2-D physical space. We describe details of analysis methods in Appendix A.6.

### 4.3 NEAR-OPTIMAL SPATIAL NAVIGATION BY DSI VECTORS

As discussed in Section 3.2, the inner product of DSI representations approximate value functions for spatial navigation. Therefore, we tested whether DSI representations actually enable near-optimal navigation in the space.

We assume that a start location (state $s_{init}$) and a goal location (state $s_G$) are randomly given in each trial such that the shortest path length is minimally 10, and an agent has to navigate between them. To solve the task, we define a vector-based state transition rule. Suppose that the agent exists at a state $s$, and a set of neighboring states of $s$ is $\mathbb{A}(s)$. Given the goal representation vector $\boldsymbol{w}(s_G)$, a value function of a neighboring state $s_{next} \in \mathbb{A}(s)$ is estimated by $\boldsymbol{x}(s_{next}) \cdot \boldsymbol{w}(s_G)$, and the agents transits to the state that has a maximum value. This state transition rule can be geometrically interpreted as the choice of movements that has the closest angle with the goal vector in the representation space (Figure 3A). Otherwise, we can interpret that the agent estimates value functions by linear readout from grid-like DSI representations. Because of the approximation error, this rule did not always give optimal navigation (the shortest path from the start to the goal). However, the agent could take the shortest path to the goal in 93.9% of 1,000 trials we tested (an example is shown in

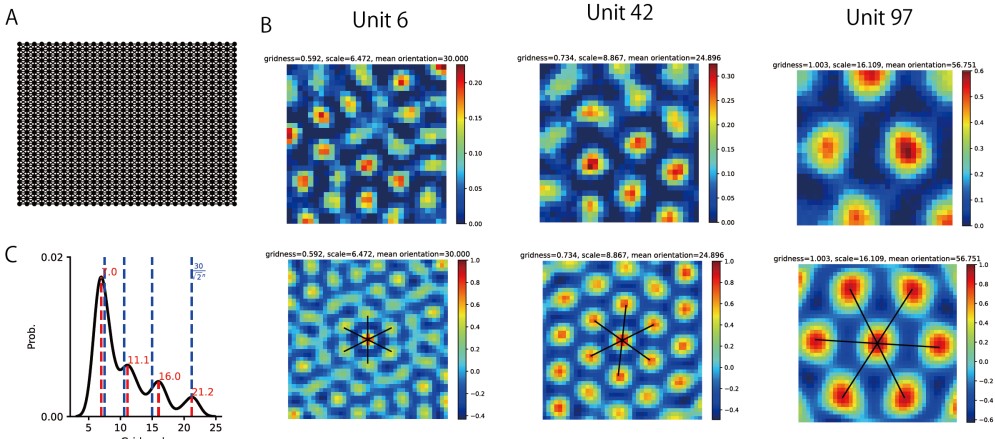

Figure 2: DSI representations for the physical space. (A) The square room tiled with 30x30 discrete states. (B) Grid-like DSI representations in the space (upper) and their spatial autocorrelation (lower). (C) The distribution of grid scales of DSI representations. Red and blue broken lines indicate positions of peaks and $\frac{30}{(\sqrt{2})^n}$, respectively.

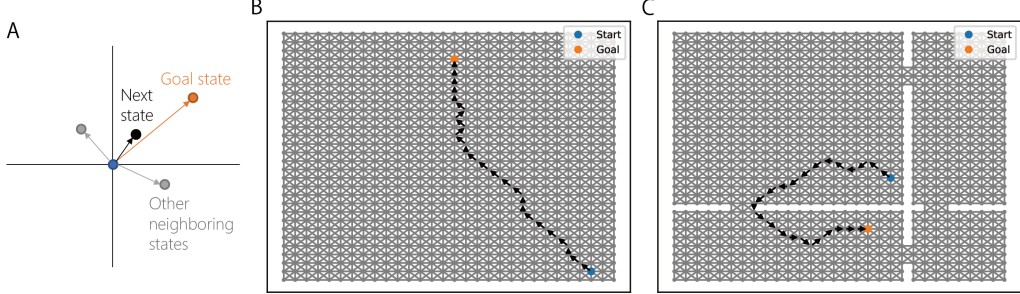

Figure 3: Spatial navigation using DSI representation vectors. (A) Geometric interpretation of the state transition rule in the DSI representation space. (B) An example spatial path obtained by DSI-based navigation. (C) Example spatial paths by DSI-based navigation in the four-room environment.

Figure 3B). Furthermore, 97.2% were near-optimal navigation in which the actual path length was shorter than 1.1 times the shortest path length. The same framework also worked in a relatively complex environment with separated rooms (Figure 3C). In this environment, the ratio of optimal and near-optimal navigation was 68% and 82.6%, respectively. We also confirmed that we can perform path integration based on DSI representations using movement-conditional recurrent weights (McNaughton et al., 2006; Burak & Fiete, 2009; Oh et al., 2015; Gao et al., 2019) (Appendix A.7). These results show that DSI representations can support spatial navigation, which corresponds to the contribution of biological grid cells for spatial navigation.

## 4.4 VECTOR-BASED INFERENCE OF SPATIAL CONTEXTS

We additionally found that we can perform vector-based inference for spatial navigation in a novel context. First, we constructed DSI representation vectors in spatial contexts A and B, each of which has a barrier (Figure 4A). Then, we created representation vectors for a novel context A+B with two barriers by simply adding representation vectors for familiar contexts A and B (Figure 4A). We tested vector-based navigation (described in the section 4.3) in three spatial contexts A, B, and A+B, using one of three representations for A, B, and A+B. Naturally, representation vectors for A and B gave the best performance in contexts A and B, respectively (Figure 4B). Notably, composite representation vectors for A+B achieved the best performance in the context A+B (Figure 4B). This

Figure 4: Vector-based inference of a novel spatial context. (A) We constructed representation vectors for a novel context A+B (a square room with two barriers) by the sum of DSI representation vectors for two familiar contexts A and B (with either of two barriers). (B) Rates of near-optimal navigation in various settings of representation vectors and contexts.

result suggests that we can utilize vector-based composition of representations for a novel spatial context. We describe details of the simulation in Appendix A.8.

Additional analysis by multidimensional scaling (MDS) suggest that summing DSI vectors leads to composition of an appropriate metric space for the novel context (Appendix A.9). This is potentially useful for composing multiple constraints that change reachability between states in various tasks (such as control of robotic arms and playing computer games), like composition of tasks in soft-Q learning (Haarnoja et al., 2018; Makino).

## 5 LEARNING REPRESENTATIONS OF CONCEPTUAL SPACES

In this section, we show that the same DSI model can learn representations for a complex conceptual space from linguistic inputs, and those representations support vector-based conceptual inference.

### 5.1 LEARNING PROCEDURE

We used text data taken from English Wikipedia, which contains 124M tokens and 9376 words (see Appendix A.10 for the detail of preprocessing). To construct DSI representations, we regarded each word as a "state", and considered the text data as a sequence of 9376 states ($N_s = 9376$). Then, we applied the exactly same learning procedure as in the experiment of physical spaces. We obtained 300-dimensional DSI representation vectors for each word. The discount factor $\gamma$ was set to 0.9. The setting of other parameters was the same as the experiment of physical spaces.

### 5.2 EMERGENCE OF CONCEPT-SPECIFIC REPRESENTATIONS

As in the previous section, we regard each dimension of representation vectors as a neural unit, and checked how various words activate those units. Specifically, we listed ten words that elicited the highest activities in each unit (TOP-10 words). Consequently, we found that many units are activated by words related to specific concepts (Figure 5; other examples in Appendix A.12), which could be named as "game cell" or "president cell", for example. We quantified this conceptual specificity through WordNet-based semantic similarity between words (Princeton University, 2010). We compared mean similarity among TOP-10 words and a null distribution of similarity between random word pairs, by which we determined statistically significant concept-specific units and quantified the degree of conceptual specificity of each unit (see Appendix A.11 for details). DSI exhibited the larger number of significantly concept-specific units and higher average conceptual specificity than other well-established word embedding methods such as skip-gram and GloVe (Table 1) (Mikolov et al., 2013a;b; Pennington et al., 2014; Levy & Goldberg, 2014). We also analyzed conceptual specificity of representations in the embedding layer of pretrained BERT model (bert-base-uncased in Hugging Face transformers) (Devlin et al., 2018; Wolf et al., 2020), which was lower than DSI (Table1). This result shows that our DSI model forms more concept-specific representations than other models.

Additional analyses revealed that word representation vectors are non-sparse and distributed (Appendix A.15). Therefore, each word is represented by the combination of concept-specific units

## TOP 10 words for each unit

| Unit 9 | Unit 31 | Unit 113 | Unit 244 | Unit 269 |
|--------|---------|----------|----------|----------|
| eleventh 2.13 | orbit 2.16 | playstation 2.38 | covid-19 1.84 | reagan 1.84 |
| tenth 2.09 | spacecraft 2.12 | nintendo 2.25 | virus 1.76 | obama 1.81 |
| ninth 2.00 | saturn 1.95 | xbox 2.13 | infected 1.71 | barack 1.78 |
| seventh 1.96 | apollo 1.84 | console 1.93 | pandemic 1.65 | bush 1.61 |
| eighth 1.95 | lunar 1.80 | arcade 1.78 | viral 1.64 | trump 1.55 |
| twelfth 1.95 | planets 1.78 | 360 1.62 | epidemic 1.62 | clinton 1.53 |
| 7th 1.93 | nasa 1.73 | gameplay 1.56 | hiv 1.50 | presidents 1.35 |
| 8th 1.86 | mars 1.73 | pc 1.46 | plague 1.42 | roosevelt 1.31 |
| 10th 1.84 | moon 1.69 | mario 1.40 | infections 1.39 | nixon 1.30 |
| 11th 1.84 | orbital 1.66 | metacritic 1.33 | outbreak 1.37 | presidential 1.29 |
| *Order cell* | *Space cell* | *Game cell* | *Virus cell* | *President cell* |

Figure 5: Concept-specific representations formed by DSI. Ten words that gave the highest activation (TOP-10 words) are shown. We also marked each unit with a descriptive label.

Table 1: Evaluation of conceptual specificity. Numbers of evaluated and significantly concept-specific units, the ratio of concept-specific units, and average conceptual specificity are summarized.

| METHOD | EVALUATED | SIGNIFICANT | RATIO | SPECIFICITY |
|--------|-----------|-------------|-------|-------------|
| DSI | 285 | 100 | 35% | 0.32 |
| Skip-gram | 297 | 14 | 5% | -0.11 |
| CBOW | 298 | 83 | 28% | 0.26 |
| GloVe | 297 | 6 | 2% | -0.07 |
| PPMI-SVD | 297 | 36 | 12% | 0.06 |
| BERT | 755 | 51 | 7% | 0.03 |

shared by several related words. For example, "France" can be represented by the combination of units which we could name French cell and country cell (Appendix A.15).

### 5.3 VECTOR-BASED COMPUTATION IN THE CONCEPTUAL SPACE

Given that DSI and word embedding methods are mathematically similar (Section A.4), we expect that DSI vectors have similar properties to representation vectors learned by those word embedding methods. We evaluated the performance of DSI vectors in two tasks that have been used to evaluate word embedding methods: word similarity and analogical inference (Mikolov et al., 2013a;b; Pennington et al., 2014; Levy & Goldberg, 2014). In the word similarity task, we calculated cosine similarity between representation vectors of word pairs, and evaluated the rank correlation between those cosine similarities and human word similarities (WS353 dataset (Agirre et al., 2009); 248/345 word pairs were used). In the analogical inference task, we performed calculation of vectors such as $x(king) - x(man) + x(woman)$ and checked whether the resultant vector has the maximum cosine similarity with $x(queen)$ (Mikolov's dataset (Mikolov et al., 2013a;b); 3157/19544 questions were used; examples in Appendix A.13). The result shows that DSI vectors achieved comparable performance with other well-established word embedding methods (Table 2).

This result indicates that similarity of DSI representation vectors corresponds to semantic similarity. This property is consistent with the experimental observation that population-level pattern similarity of concept cell activities represents semantic categories (Reber et al., 2019). By visualizing the structure of DSI representations by MDS, we can actually see clustering of words corresponding to 10 semantic categories used in Reber et al. (2019) (Appendix A.14). Furthermore, conceptual inference is possible through arithmetic composition of DSI vectors.We additionally found that this inference is intuitive recombination of concept-specific units in some cases. For example, transformation from "Paris" to "France" corresponds to activation of country cell and deactivation of capital cell, which is possible by summing the difference of Germany and Berlin vectors. (Appendix A.15).

Table 2: Performances of vector-based computations in the conceptual space. The correlation of word similarity evaluated by vectors and humans, and the rate of correct analogical inference.

| METHOD | SIMILARITY | ANALOGY |
|---|---|---|
| DSI | 0.70 | 0.63 |
| Skip-gram | 0.69 | 0.71 |
| CBOW | 0.69 | 0.74 |
| GloVe | 0.69 | 0.71 |
| PPMI-SVD | 0.52 | 0.58 |

## 6 DISCUSSION

In this paper, we proposed a theoretically interpretable and biologically plausible neural representation model for physical and conceptual spaces. We demonstrated that our DSI model forms grid-like representations in the physical space and concept-specific representations in the linguistic space, which are assumed to correspond to neural representations in EC. Furthermore, we showed that SI is mathematically related to linear reinforcement learning and word embedding methods, thus DSI representations support spatial navigation and conceptual inference. These results suggest that we can extend the spatial representation model of EC to learn and compute linguistic concepts, which apparently seems a different computational domain from physical space.

In the section 5.2, we demonstrated concept-specific representations created from text data. To the best of our knowledge, such property has not been reported in any word embedding methods. However, we unexpectedly found that continuous-bag-of-words (CBOW) showed relatively high conceptual specificity. Although DSI is related to PMI, skip-gram and GloVe, we have not found relationship to CBOW. Further clarification of necessary conditions for conceptual specificity is still open problem.

Although DSI has clear mathematical interpretation, how biological neural networks can learn DSI is still unclear. A possible solution is an extension of skip-gram neural network with SR, non-negativity, and decorrelation. Because SI corresponds to PMI which is the optimum of skip-gram neural network, we can expect that extended skip-gram network learns DSI. Building such model and relating it to the circuit mechanism in hippocampus and EC are left for future research.

Our model relates word embedding to conceptual representations in the brain. Previously study showed that skip-gram representations support high-performance decoding of semantic information from fMRI data (Nishida & Nishimoto, 2018). Another study revealed that hippocampal theta oscillation codes semantic distances between words measured in word2vec subspace (Solomon et al., 2019). These experimental results support our hypothesis. However, recent studies have shown that representations in transformer-based models (Vaswani et al., 2017) such as GPT (Brown et al., 2020) achieve remarkable performance in linear fitting to neural recording during linguistic processing (Goldstein et al., 2022; Schrimpf et al., 2021). A major difference between our DSI model and transformer-based models is that DSI representations are basically fixed (static embedding) whereas transformer-based models flexibly create context-dependent representations (dynamic embedding). Conceptual interpretation obviously depends on the context, thus activities of concept cells are context-dependent (Bausch et al., 2021). Therefore, our DSI model should be extended to process context-dependence, hopefully by combination with other models for learning context-dependent latent cognitive states (Uria et al., 2020; George et al., 2021; Whittington et al., 2020).

Another direction of future research is application to general conceptual spaces by learning DSI representations from low-level sensory inputs, like spatial learning from visual and auditory inputs in previous models (Banino et al., 2018; Taniguchi et al., 2018; Uria et al., 2020). It may be possible by learning discrete states by unsupervised clustering for deep networks (Caron et al., 2018). As for the human brain, infants probably form primitive spatial and conceptual representations from sensory signals, and later linguistic inputs enrich those representations. We speculate that real-world sensory data also contain the information of the conceptual space, for which DSI can be extended to learn those structures. Such model would clarify the role of the hippocampal system in computation of general conceptual spaces.

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

## A APPENDIX

### A.1 CALCULATION OF SR

SR is a variant of value functions in reinforcement learning, thus we can use various methods such as temporal-difference (TD) learning for the construction. Throughout this study, we used a direct count method because we performed only offline processing of finite data. In a sequence of states $\{s_1, \ldots, s_t, \ldots, s_T\}$, we recursively calculated exponential traces of past states $z(s, t) = \sum_{\tau=0}^{t-1} \gamma^\tau \delta(s_{t-\tau}, s)$ as

$$z(s, t) = \gamma z(s, t-1) + \delta(s_t, s), \tag{11}$$

and calculated SR from state counts and coincidence counts as

$$SR(s, s') = \frac{\sum_{t=1}^{T} z(s, t)\delta(s_t, s')}{\sum_{t=1}^{T} \delta(s_t, s)}. \tag{12}$$

### A.2 DETAILS OF DECORRELATIVE NMF

In decorrelative NMF, we iteratively updated vectors $\boldsymbol{x}(s)$ and $\boldsymbol{w}(s)$ by Nesterov's accelerated gradient descent method to minimize the objective function (Eq. 4), rectifying all elements every iteration. Gradients are

$$\frac{\partial J}{\partial x_k(s)} = -\sum_{s'} \rho(s, s')(PSI(s, s') - \boldsymbol{x}(s) \cdot \boldsymbol{w}(s'))w_k(s')$$

$$+ \beta_{cor} \sum_{j \neq k} \frac{Corr(k, j)\tilde{x}_j(s)}{\sqrt{\sum_s(\tilde{x}_k(s))^2 \sum_s(\tilde{x}_j(s))^2}} + \beta_{reg}x_k(s), \tag{13}$$

$$\frac{\partial J}{\partial w_k(s')} = -\sum_s PSI(s, s')(PSI(s, s') - \boldsymbol{x}(s) \cdot \boldsymbol{w}(s'))x_k(s) + \beta_{reg}w_k(s'). \tag{14}$$

We note that we regarded mean and variance of $x_i(s)$ in the correlation ($\frac{1}{N_s}\sum_s x_i(s), \sum_s(\tilde{x}_i(s))^2$ in Eq. 6) as constants in the calculation of these gradients. Practically, this heuristic did not affect the performance of decorrelation.

Throughout this paper, the learning rate was 0.05 and the number of iteration was 10000. Parameters were $\beta_{cor} = 1, \beta_{reg} = 0.001$, and $\rho_{min} = 0.001$.

### A.3 MATHEMATICAL RELATIONSHIP OF DSI AND REINFORCEMENT LEARNING

In this section, we show that our model approximates value functions of linear reinforcement learning (Todorov, 2006; 2009; Piray & Daw, 2021) in the setting of spatial navigation.

In linear reinforcement learning, an agent aims to maximize "gain" instead of reward. Assuming a default policy $\pi^d(s)$ (any policy is available; typically random walk in the case of exploration task), gain function is defined as

$$g(s) = r(s) - \lambda KL(\pi(s)|\pi^d(s)), \tag{15}$$

where $r(s)$ is expected reward at the state $s$ and $\lambda KL(\pi(s)|\pi^d(s))$ is the cost imposed on the difference between the current policy $\pi(s)$ and the default policy $\pi^d(s)$ ($\lambda$ is a relative weight of the cost). Then, previous works have shown that the optimal policy and corresponding value functions can be determined explicitly by solving linear equations (Todorov, 2006; 2009; Piray & Daw, 2021). Here we consider an environment that consists of $N_N$ non-terminal states and $N_T$ terminal states. We define two transition probability matrices under the default policy: $\boldsymbol{P}_{NT}$ is a $N_N \times N_T$ matrix for

transitions from non-terminal states to terminal states, and $\boldsymbol{P}_{NN}$ is a $N_N \times N_N$ matrix for transitions across non-terminal states. Furthermore, $\boldsymbol{r}_N$ and $\boldsymbol{r}_T$ are vectors of rewards at non-terminal states and terminal states, respectively. In this condition, a vector of value functions under optimal policy $\boldsymbol{v}^* = (v^*(s_1), \ldots, v^*(s_{N_N}))$ is obtained as

$$\exp(\lambda^{-1}\boldsymbol{v}^*) = \boldsymbol{M}\boldsymbol{P}_{NT}\exp(\lambda^{-1}\boldsymbol{r}_T), \tag{16}$$

where $\boldsymbol{M} = (\mathrm{diag}(\exp(-\lambda^{-1}\boldsymbol{r}_N)) - \boldsymbol{P}_{NN})^{-1}$ is DR (Piray & Daw, 2021).

To relate $\boldsymbol{v}^*$ to SI, we consider a specific condition in which the environment consists of non-terminal states, and a virtual terminal state is attached to a goal state $s_G$ arbitrarily chosen from non-terminal states (Figure 1B). When the agent gets to the goal, it transits to the terminal state with a probability $p_{NT}$. Furthermore, we assume that reward at non-terminal states are uniformly negative and reward at the terminal state is positive so that the agent has to take a short path to goal to maximize reward. Specifically, we assume all elements of $\boldsymbol{r}_N$ are $\lambda \log \gamma$, and $r_T = -\lambda(\log \gamma + \log p_{NT} + \log P^d(s_G))$ where $\gamma$ is an arbitrary value in the range $(0,1)$, and $P^d(s_G)$ is a probability of visiting the state $s_G$ under the default policy. Then, we obtain

$$\exp(\lambda^{-1}\boldsymbol{v}^*) = \frac{1}{P^d(s_G)}(\boldsymbol{I} - \gamma P_{NN})^{-1}\boldsymbol{e}^{(i_G)}, \tag{17}$$

where $\boldsymbol{e}^{(i_G)} = (0, \ldots, 0, 1, 0, \ldots, 0)^T$ ($i_G$ is the index of the goal state). Because $(\boldsymbol{I} - \gamma P_{NN})^{-1}$ is equivalent to a successor representation matrix with a discount factor $\gamma$ (Dayan, 1993; Stachenfeld et al., 2017), we finally obtain

$$\lambda^{-1}v^*(s) = \log(SR^d(s, s_G)) - \log P^d(s_G) = SI^d(s, s_G) \approx \boldsymbol{x}(s) \cdot \boldsymbol{w}(s_G), \tag{18}$$

where $SR^d(s, s_G)$ and $SI^d(s, s_G)$ are SR and SI under the default policy, respectively. Thus, SI is proportional to value functions for spatial navigation and inner products of DSI vectors approximates value functions. Based on this interpretation, we basically regard $\boldsymbol{x}(s)$ as a representation of each state, and $\boldsymbol{w}(s)$ represents a temporary goal.

## A.4 MATHEMATICAL RELATIONSHIP OF DSI AND WORD EMBEDDING

In this section, we discuss the relationship of SI and PMI (Levy & Goldberg, 2014) in detail. PMI is

$$PMI = \log\left(\frac{P(word_i, word_j)}{P(word_i)P(word_j)}\right), \tag{19}$$

where $P(word_i, word_j)$ is a coincidence probability of two words (in a certain temporal window).

To relate PMI to SI, we regard words as states: $s = word_i, s' = word_j$. Furthermore, we consider a specific way to count coincidence probability. In typical word embedding, a finite symmetric rectangular window is often used:

$$P(s, s') = \sum_{t=0}^{W} P(s_t = s', s_0 = s), \tag{20}$$

where $W$ is a window size. Here, we implicitly assumed that same state (word) is not repeated in the temporal window to guarantee that P(s, s') is probability.

However, we may arbitrarily calculate coincidence for $P(s, s')$. Here we evaluate coincidence with an infinite asymmetric exponential kernel as in SR:

$$P(s, s') = (1 - \gamma)\sum_{t=0}^{\infty} \gamma^t P(s_t = s', s_0 = s). \tag{21}$$

We introduced a normalization factor $(1 - \gamma)$ to guarantee that $P(s, s')$ is less than one $((1 - \gamma)\sum_{t=0}^{\infty}\gamma^t = 1)$. Then, PMI becomes

$$PMI = \log\left(\frac{(1-\gamma)\sum_{t=0}^{\infty}\gamma^t P(s_t = s', s_0 = s)}{P(s)P(s')}\right) \tag{22}$$

$$= \log(SR(s, s')) - \log(P(s')) + \log(1 - \gamma) \tag{23}$$

$$= SI(s, s') + \log(1 - \gamma). \tag{24}$$

If we perform dimension reduction, $\log(1 - \gamma)$ can be ignored because it is a constant. Therefore, we can interpret SI as a special case of PMI in our model.

## A.5 RELATIONSHIP BETWEEN MODEL COMPONENTS AND REPRESENTATIONS

To clarify the contribution of each model component to the results in this study, we performed a "lesion study" in which we removed some components in DSI and repeated the same evaluation procedure in the main text. We summarize results in Table 3. First, we tested representations obtained by singular value decomposition of successor representation (SR-SVD), which was regarded as a model of grid cells in a previous study (Stachenfeld et al., 2017). DSI model exceeded SR-SVD in all aspects shown in this study. Next, we tested DSI model without decorrelation ($\beta_{cor} = 0$) and DSI model without non-negativity (no rectification of representation vectors). Neither modification impaired the performance of navigation and inference, showing the importance of using SI for vector-based computations as theoretically expected. In contrast, removing decorrelation and non-negativity significantly impaired the emergence of grid-like units and concept-specific units, respectively. Thus, decorrelative NMF is crucial to obtain biologically plausible representations.

Table 3: Contribution of each model component to the performance. Rates of grid-like units, near-optimal navigation, significantly concept-specific units, correct analogical inference.

| METHOD | GRID CELL | NAVIGATION | CONCEPT CELL | ANALOGY |
|---|---|---|---|---|
| Full DSI | 51% | 97.2% | 35% | 63% |
| Decorrelation OFF | 5% | 99.9% | 38% | 64% |
| Non-negativity OFF | 29% | 99.8% | 2% | 63% |
| SR-SVD | 21% | 57.2% | 13% | 28% |

## A.6 DETAILS OF EVALUATION OF GRID REPRESENTATIONS

In the section 4.2, we performed the gridness analysis following a previous experimental study (Sargolini et al., 2006). For each unit, we rotated the spatial autocorrelation map (Figure 2B, lower) and calculated correlations between the original and rotated maps. Gridness was defined as the difference between the lowest correlation at $60°$ and $120°$ and the highest correlation at $30°$, $90°$ and $150°$. A unit was classified as a grid cell when gridness exceeds zero.

In Figure 2C, we constructed a distribution of grid scales. Grid scales were determined as the median of distances between the central peak and the six closest peaks (vertices of inner hexagon) in the spatial autocorrelation map. The kernel function for kernel density estimation was Gaussian with a standard deviation 1.

## A.7 PATH INTEGRATION BY DSI

We performed path integration based on DSI representations using movement-conditional recurrent weights. This strategy has been used in previous studies such as grid cell modeling (Gao et al., 2019) and action-conditional video prediction (Oh et al., 2015). This mechanism is also consistent with a conventional biological model for path integration in which head direction signals activate one of attractor networks specialized for different directional shifts of grid patterns (McNaughton et al., 2006; Burak & Fiete, 2009).

We made an estimate of the next representation vector $\hat{x}_{t+1}$ by linear transformation of the current representation vector $x(s_t)$ as

$$\hat{x}_{t+1} = M(a_t)x(s_t) \tag{25}$$

where $a_t$ represents a movement (one of eight directional movements in this study) and $M(a_t)$ is movement-conditional recurrent weight matrix. Here, $x(s_t)$ was a DSI representation vector, and we optimized the matrix $M(a_t)$ by minimizing prediction error $||x(s_{t+1}) - M(a_t)x(s_t)||_2^2$ by stochastic gradient descent during random walk on the state transition graph (20 simulation trials of 100,000 time steps). After optimization, we set an initial state $s_0$ and a sequence of movements $\{a_0, a_1, \ldots, a_{T-1}\}$, and performed path integration by recursive estimation $\hat{x}_{t+1} = M(a_t)\hat{x}_t$. We determined a position at each time step by searching a state representation vector that has minimum Euclidian distance with the estimated vector ($s_t = \arg\min_s ||x(s) - \hat{x}_t||_2$). As shown in Figure 6, this strategy gave accurate estimation of the spatial path from movement signals.

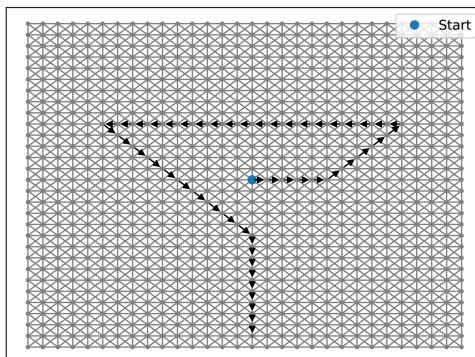

Figure 6: Path integration using DSI model. (Left) Actual path. (Right) Path estimated from DSI vectors updated by movement information.

## A.8 DETAILS OF VECTOR-BASED INFERENCE OF THE SPATIAL CONTEXT

In the section 4.4, we performed vector-based inference for spatial navigation in a novel context. Specifically, we define separated states in contexts A, B, A+B as $s_i^A, s_i^B$ and $s_i^{A+B}$, where $i$ is a positional index which indicates a same position in all contexts ($i = 1, 2, \cdots, 900$). We constructed representation vectors $\boldsymbol{x}(s_i^A)$, $\boldsymbol{w}(s_i^A)$, $\boldsymbol{x}(s_i^B)$, and $\boldsymbol{w}(s_i^B)$ through direct experiences, then we created $\boldsymbol{x}(s_i^{A+B})$ and $\boldsymbol{w}(s_i^{A+B})$ as

$$\boldsymbol{x}(s_i^{A+B}) = \boldsymbol{x}(s_i^A) + \boldsymbol{x}(s_i^B), \tag{26}$$

$$\boldsymbol{w}(s_i^{A+B}) = \boldsymbol{w}(s_i^A) + \boldsymbol{w}(s_i^B). \tag{27}$$

We performed spatial navigation in a given context using one of three representations $\{\boldsymbol{x}(s_i^A), \boldsymbol{w}(s_i^A)\}$, $\{\boldsymbol{x}(s_i^B), \boldsymbol{w}(s_i^B)\}$, and $\{\boldsymbol{x}(s_i^{A+B}), \boldsymbol{w}(s_i^{A+B})\}$ for corresponding positions, following the rule described in the section 4.3. In figure 7, we show structures of state transition graphs for three contexts A, B, and A+B.

To learn representations in contexts A and B, we sampled sequences of $\{s_i^A\}_{i=1,\cdots,900}$ and $\{s_i^B\}_{i=1,\cdots,900}$ by random walk in context A and B. The procedure was basically same with the section 4 except that we increased the number of simulation trials from 500 to 1,000, and state transition to the same position in the other context occurred every 5,000 time steps (transition between $s_i^A$ and $s_i^B$). We added this transition to associate the same position in different contexts. It means that we assumed that the setting of barriers can change during the experience but this temporal association may be substituted by similarity of sensory inputs across contexts. From sampled sequences, we calculated PSI for all combinations of $\{s_i^A\}_{i=1,\cdots,900}$ and $\{s_i^B\}_{i=1,\cdots,900}$, and calculated 100-dimensional DSI vectors for 1,800 states by simultaneous compression of all states. The discount factor $\gamma$ was set to 0.999.

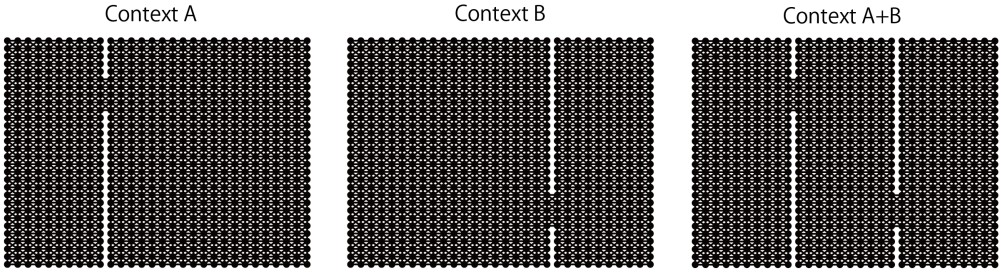

Figure 7: Structures of state transition graphs used in the section 4.4.

## A.9 Visualization of spatial structures represented by DSI vectors

In Figure A.9, we visualized metric spaces defined by representation vectors for contexts A and B, and composite vectors for the context A+B, by using multidimensional scaling (MDS). This visualization clearly shows that DSI vectors for A and B capture structures of spatial contexts A and B, and adding those vectors yields appropriate metric space for novel context A+B.

Representation vector A    Representation vector B    Composite vector A+B

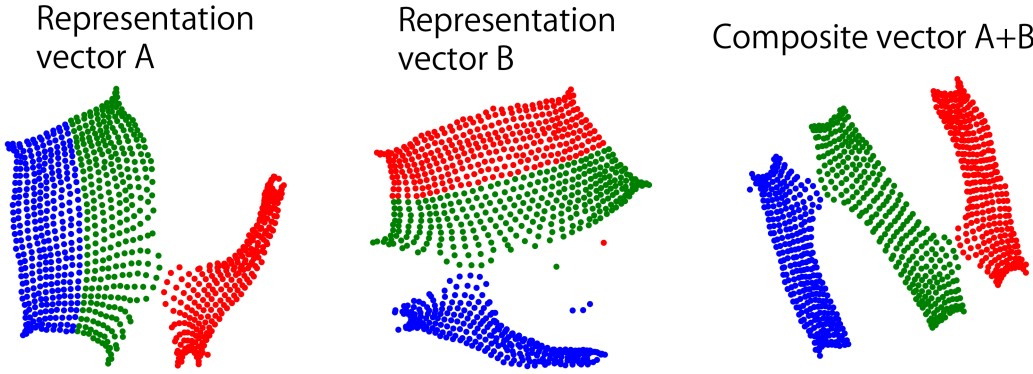

Figure 8: Visualization of metric spaces defined by representation vectors for contexts A and B, and composite vectors for the context A+B using MDS. Red, blue, and green dots correspond to states (positions) in the left, center, and right parts of the original 2-D space (Figure 7), respectively.

## A.10 Details of preprocessing of text data

In the section 5, we used text data taken from English Wikipedia dump (enwiki-latest-pages-articles, 22-May-2020). We first generated text files from raw data using wikiextrator (`https://github.com/attardi/wikiextractor`). We tokenized texts by nltk Punkt sentense tokenizer, and randomly sampled 100,000 articles containing 1,000 tokens at minimum. We lower-cased all characters and removed punctuation characters in the data. After that, we selected words that appeared more than 1,000 times in the data, and substituted all other rare words by <unk> symbol. Finally, we obtained data that contains 124M tokens and 9376 words.

## A.11 Details of evaluation of conceptual specificity

In the section 5.2, conceptual specificity of each unit was evaluated using WordNet database (Princeton University, 2010). In WordNet, a word belongs to several synsets (sets of cognitive synonyms), and semantic similarity of two synsets can be evaluated from the shortest path length between them in the WordNet structure (we used path_similarity function in nltk library). We defined similarity of two words as the highest similarity among all combinations of synsets of those words. We calculated mean similarity of all combinations of TOP-10 words (ten words that highly activated the unit; Figure 5A) that are available in WordNet. We evaluated only units which had at least five TOP-10 words available in WordNet. Furthermore, we randomly generated 1,000 pairs of words available in WordNet, and generated a null distribution of similarity between words. We defined a significance threshold of similarity as a 95 percentile of the null distribution, and a unit was classified as a significantly concept-specific unit if mean similarity of TOP-10 words exceeded the threshold. Furthermore, we quantitatively defined a conceptual specificity of each unit as

$$\frac{s_{unit}}{s_{null}} - 1, \tag{28}$$

where $s_{unit}$ is mean similarity of TOP-10 words and $s_{null}$ is the mean of the null distribution. This quantity becomes zero if similarity between TOP-10 words is not different from random pairs, and becomes positive if TOP-10 words are semantically similar. This conceptual specificity was averaged over all evaluated units.

A.12 EXAMPLE DSI REPRESENTATIONS FOR WORDS

In the figure 9, we show TOP-10 words of DSI units without manual selection. We found several non-significant units exhibit conceptual specificity according to manual inspection (for example, unit4 may be named as university cell). This is probably because of the limitation of knowledge covered by WordNet. Therefore, we suppose that the current evaluation method tends to underestimate the number of concept-specific units. However, the comparison across models was fair because we used the same procedure and criteria for all models.

| Unit 1 | Unit 2 | Unit 3 | Unit 4 | Unit 5 |
|---|---|---|---|---|
| faced | transit | humid | universities | sheets |
| overcome | commuter | winters | colleges | olivia |
| facing | trains | summers | uc | compact |
| due | bus | climate | usc | senators |
| owing | buses | mild | consortium | antony |
| citing | subway | precipitation | institutes | proposes |
| experiencing | rail | warm | carnegie | atlas |
| cope | passenger | temperatures | affiliated | spaces |
| amid | metro | cool | alumni | geometry |
| suffered | passengers | cold | selective | rough |

| Unit 6 | Unit 7 | Unit 8 | Unit 9 | Unit 10 |
|---|---|---|---|---|
| designers | rebounds | nbc | eleventh | nhl |
| scientists | assists | aired | tenth | confirmation |
| experts | averaged | abc | ninth | hockey |
| professionals | steals | airing | seventh | maple |
| consumers | nba | cbs | eighth | roller |
| researchers | averaging | syndicated | twelfth | judiciary |
| filmmakers | partition | programming | 7th | flames |
| lawyers | points | broadcast | 8th | devils |
| composers | prussia | channel | 10th | batch |
| scholars | mvp | espn | 11th | senators |

| Unit 11 | Unit 12 | Unit 13 | Unit 14 | Unit 15 |
|---|---|---|---|---|
| lowest | valve | archive | hatch | rotten |
| ranked | wheel | archives | deaf | tomatoes |
| highest | steering | manuscripts | robot | metacritic |
| ranks | rotating | collections | suffrage | aggregate |
| literacy | wheels | collection | egg | approval |
| demographic | cylinder | manuscript | seymour | reviews |
| rising | trigger | library | salem | grossed |
| prevalence | shaft | catalogue | wwe | rating |
| ratings | pump | valuable | monte | byron |
| rank | gear | documents | resist | apparatus |

Figure 9: A part of word representations formed by DSI. Ten words that gave the highest activation (TOP-10 words) are shown for units 1 - 15. Those classified as concept-specific units in our analysis are boxed with bold lines.

## A.13 EXAMPLES OF THE ANALOGICAL INFERENCE TASK

In table 4, we show some examples of the analogical inference in Mikolov's dataset. There is a relationship "WORD1 is to WORD2 as WORD3 is to WORD4". Then, an expected relationship in the vector space is WORD2-WORD1=WORD4-WORD3. In this study, we performed inference of WORD4 by WORD3+WORD2-WORD1. We regarded an inference was correct if the actual vector of WORD4 had the largest cosine similarity to the inferred vector among all word representation vectors (except those for WORD1, WORD2, and WORD3). If the number of words is 10,000, a chance level of the correct answer rate is 0.01%. Therefore, the performance shown in this study (more than 50%) is far above the chance level.

Table 4: Examples of analogical relationships in Mikolov's dataset.

| WORD1 | WORD2 | WORD3 | WORD4 |
|---|---|---|---|
| boy | girl | king | queen |
| husband | wife | he | she |
| Paris | France | Tokyo | Japan |
| China | Chinese | Italy | Italian |
| big | bigger | small | smaller |
| free | freely | quick | quickly |
| convenient | inconvenient | possible | impossible |

## A.14 CLUSTERING OF SEMANTIC CATEGORIES IN DSI SPACE

Figure 10 shows the structure of DSI word representations visualized by MDS. We arbitrarily chose words based on 10 semantic categories used in Reber et al. (2019). We used same dissimilarity metric with Reber et al. (2019) (1 - Pearson's correlation coefficient).

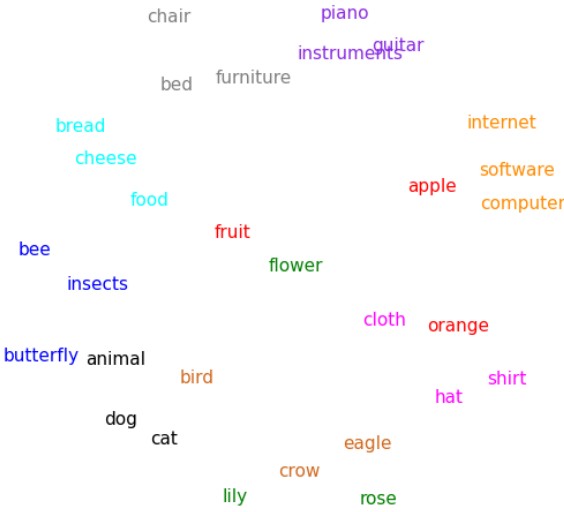

Figure 10: Visualization of the representational structure of DSI using MDS. Words are plotted in coordinates embedded in 2-D DSI space. Each color corresponds to a semantic category used in Reber et al. (2019).

A.15    INTUITIVE MECHANISM OF WORD REPRESENTATIONS BY DSI

In this section, we discuss how DSI vectors represent and compute words.

First, we analyzed the ratio of each element to the sum of all elements in DSI vectors. We found that even the largest element accounted for 5% of the sum of all elements on average. (Figure 11). This result shows that DSI vectors for words are non-sparse and distributed, thus each word is represented by the combination of multiple conceptual units.

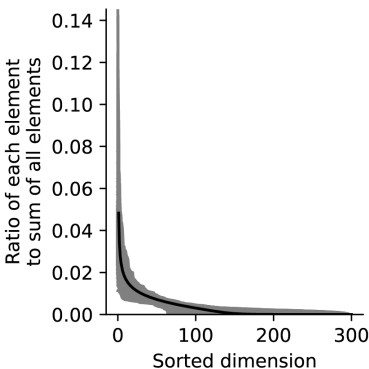

Figure 11: Non-sparsity of word representations. Gray lines show ratio of each element to sum of all elements in each word representation vector, and the black line is the average of them. Elements were sorted in descending order.

The most active units in each vector (the largest absolute values)

| France | Germany | Paris | Berlin |
|---|---|---|---|
| 1.    Unit 285 | 1.    Unit 285 | 1.    Unit 142 | 1.    Unit 134 |
| 2.    Unit 142 | 2.    Unit 134 | 2.    Unit 281 | 2.    Unit 281 |

| France-Paris | Germany-Berlin | France-Germany | Paris-Berlin |
|---|---|---|---|
| 1.    Unit 281 | 1.    Unit 285 | 1.    Unit 142 | 1.    Unit 142 |
| 2.    Unit 285 | 2.    Unit 281 | 2.    Unit 134 | 2.    Unit 134 |

TOP 10 words for each unit

| Unit 134 | Unit 142 | Unit 281 | Unit 285 |
|---|---|---|---|
| friedrich | et | stockholm | belgium |
| heinrich | des | budapest | finland |
| ludwig | le | oslo | bulgaria |
| karl | du | frankfurt | denmark |
| johann | les | são | countries |
| von | françois | copenhagen | greece |
| und | jacques | istanbul | colombia |
| ernst | michel | paulo | switzerland |
| wilhelm | pays | amsterdam | austria |
| der | jean | seoul | croatia |
| *German cell* | *French cell* | *Capital cell* | *Country cell* |

Figure 12: Words as combination of concepts and inference as recombination. (Upper) The most active units (the largest absolute values) in representation vectors for France, Germany, Paris, Berlin, and difference vectors between them. (Lower) TOP-10 words of active units and their interpretation.

Next, for further clarification, we inspected representations of an example set of words: France, Paris, Germany and Berlin. We can see there are two analogical relationships (country-capital and French-German relationships). We identified the most active units (TOP-2) in DSI vectors for those words, and listed TOP-10 words for identified units. As a result, we could see that "France" is represented by the combination of units that we could name as French cell and country cell, whereas

"Berlin" is represented by the combination of German cell and capital cell, and so on (Figure 12). This example also gives a simple interpretation of word similarity in DSI vector space. If words are similar, they share large number of active units, like the country cell shared by representations of France and Germany. Thus, semantic similarity between words increases cosine similarity between word vectors.

Furthermore, we also identified the largest elements (the largest absolute values) in the difference vectors between words, and found that they correspond to semantic difference between words (Figure 12). Thus, we can regard analogical inference by DSI vectors as recombination of conceptual units. For example, adding Germany-Berlin vector to Paris vector deactivate capital cell and activate country cell, which leads to the transformation of Paris into France.

Such property of the vector space is same as conventional word embedding methods, but unique feature of our model is that those analogical relationships are factorized into separated units. We speculate that constraints of decorrelative NMF are sufficient conditions to align each semantic factors to each axis of the word vector space, and the mechanism is probably related to how disentangled representations emerge in visual feature learning model (Higgins et al., 2017; Carbonneau et al., 2020).

