# OpenReview forum: "Unified neural representation model for physical and conceptual spaces"
_ICLR.cc/2023/Conference — Submitted to ICLR 2023_

### Official Review · Reviewer_pkWT · 2022-10-23

**Confidence:** 4
**Correctness:** 4
**Technical Novelty And Significance:** 4
**Empirical Novelty And Significance:** 4
**Recommendation:** 8

**Clarity, Quality, Novelty And Reproducibility:**

The paper is very clearly written, and novel. I'm fairly confident I could reproduce the results (but see my Q3 above).

**Strength And Weaknesses:**

Strengths and weaknesses

Strengths: The paper suggests a profound connection between two superficially different domains of representation - physical and conceptual space. As well as shedding light on the neuroscientific question of how the hippocampal formation functions, the authors’ method potentially provides a powerful method for use in ML. The method is well motivated, both scientifically and mathematically.

Weaknesses: I don’t see any important weaknesses. But I have some questions.

Q1. Section 4.2: Emergence of grid-like representations. You present results for x(s), but I was wondering whether grid-like representations emerge in w(s) too? In the objective function (Eq.6), x(s) and w(s) are treated almost symmetrically, but the correlation term only applies to x(s) not to w(s) (Eq.8). Is this constraint responsible for the grid-like representations?

Q2. Is there an intuitive meaning to w(s) in the word embedding case? It seems that all the results presented for conceptual space use x(s) only. (Is that correct?)

Q3. I’m a little unclear on how the optimisation process works. If x and w are functions that map state vectors to vectors with reduced dimensionality then how are they represented. My reading is they are represented in tabular fashion, i.e. x(s) and w(s’) are represented independently for all s and s’. Is that right? I assume x and s could be approximated by a neural net, but that isn’t the method used here. Is that correct?

**Summary Of The Paper:**

The paper presents a model of hippocampal / entorhinal representation that handles both physical and conceptual space with a common mathematical formulation, drawing on successor representations from RL, and building on dimensionality reduction accounts of grid cells such as the work of Dordek et al. The authors propose a novel dimensionality reduction technique that generates a disentangled pair of vectors. In the case of physical space, the authors show that grid cell-like representations emerge with their method. The authors show that their representations support spatial navigation, where one of the two vectors represents the starting position and the other represents the goal position. The most striking finding of the paper is that the very same method can be used in a language setting to generate word embeddings. The resulting word embeddings exhibit desirable properties familiar from the literature such as conceptual clustering and the ability to support analogical inference through vector algebra.

**Summary Of The Review:**

The paper is excellent, in my opinion. The proposed method is novel and makes some quite deep connections that could have important implications. The paper easily merits acceptance at ICLR, unless one of the other reviewers uncovers a serious shortcoming.

---

> ### Author Response · Authors · 2022-11-18
> **Response to Reviewer pkWT**
>
> We appreciate constructive and helpful comments from the reviewer. Here we provide point-to-point reply to the comments.
>
> Comment: Q1. Section 4.2: Emergence of grid-like representations. You present results for x(s), but I was wondering whether grid-like representations emerge in w(s) too? In the objective function (Eq.6), x(s) and w(s) are treated almost symmetrically, but the correlation term only applies to x(s) not to w(s) (Eq.8). Is this constraint responsible for the grid-like representations?
>
> Reply: As the reviewer pointed out, w(s) also becomes grid-like in the current setting. We described results of gridness analysis of w(s) in Section 4.2. We initially tested constraints on both x(s) and w(s), but we empirically found that either was enough, and less constraint was better for stability of learning. In our experience, constraints on x(s) or w(s) did not make significant difference probably because state transition structure is symmetric in the current setting. If transitions are highly asymmetric, that difference may be important.
>
> Comment: Q2. Is there an intuitive meaning to w(s) in the word embedding case? It seems that all the results presented for conceptual space use x(s) only. (Is that correct?)
>
> Reply: As the reviewer pointed out, we showed results about x(s) only. We also evaluated the performance of w(s), which was similar but slightly worse than x(s). Intuitively, x(s) and w(s) represents a current word and a future word, respectively, because s is the current state and s’ is the future state in SR(s, s’). However, further functional implication of those vectors in linguistic processing is still difficult, thus it is also an important topic in future study.
>
> Comment: Q3. I’m a little unclear on how the optimisation process works. If x and w are functions that map state vectors to vectors with reduced dimensionality then how are they represented. My reading is they are represented in tabular fashion, i.e. x(s) and w(s’) are represented independently for all s and s’. Is that right? I assume x and s could be approximated by a neural net, but that isn’t the method used here. Is that correct?
>
> Reply: As the reviewer figured out, we independently defined vectors x(s), w(s) for all discrete states s, and set random initial values, then we directly updated elements in vectors by gradient descent (practically we calculated x and w in the matrix form, using numpy and cupy). Because it is gradient descent, we may be able to use neural networks to learn representations in principle. However, we would like to leave such extension for future study.

---

> > ### Comment · Reviewer_pkWT · 2022-11-21
> > **Response to authors**
> >
> > Thanks to the authors for their replies to my queries. I think it's a good paper, and I hope it gets in.

---

> > > ### Author Response · Authors · 2022-12-09
> > > **Thanks for reply**
> > >
> > > Thanks for your encouragement!

---

### Official Review · Reviewer_33UN · 2022-10-23

**Confidence:** 3
**Correctness:** 2
**Technical Novelty And Significance:** 3
**Empirical Novelty And Significance:** 2
**Recommendation:** 3

**Clarity, Quality, Novelty And Reproducibility:**

Overall, I think there are some novel ideas in the paper (like de-correlative NMF), and the writing is quite clear. However, the results don't really support the claims made by the authors.


**Strength And Weaknesses:**

Overall I think this is an interesting paper but I'm not sure if ICLR is the right venue for this work. This work mainly tries to establish a connection between their approach (disentangled successor information) and perceptual and conceptual processing in the brain. This is an interesting and valuable endeavor but it's unclear how relevant this would be to the ICLR community. To me it seems like a cogsci/neuro venue or even NeurIPS would be a better choice for this work.

In general, I think the paper is well written and easy to follow.

In terms of contributions, the additional step of using decorrelative NMF to get state representations seems novel. However, the connection of successor representations to value functions is well-known and the connection between DSI and value functions seem to follow from this, so the connection is not quite novel. Perhaps the connection to word representations in skip-gram models is novel but given that successor representations are essentially bi-gram models, this is not very surprising.

In terms of results, there is a nice evaluation of the technique on spatial navigation and on word processing but these seem rather limited. Finding that the learned representations resemble grid cells is interesting but not a very strong evidence since many other techniques also learn representations with grid-like receptive fields.

Similary for word embedding results, I didn't really find them very strong. The authors show that features in their learned representations are on average more concept specific than some alternative techniqes. Since the brain also seems to have concept specific cells, they take this as evidence for brain using a similar mechanism. However, as far as I know it is still not very clear to what extent the concept representations in the brain are like concept cells. There seems to be many cells that do not cleanly respond to a specific concept for example. Even if this was not an issue (and we knew brain used concept cells), it is still not very strong evidence because other techniques also learn representations with similar properties. In fact, one could also directly use one-hot vectors as word representations and these would be 100% concept specific.

Overall, I think it'd be great to have more evidence supporting the use of representations like DSI in the brain, by perhaps testing out other properties of these representations (like word similarity etc.).

One major issue with the paper for me is that it conflates conceptual processing with language processing. Language (and words) are only a part of conceptual processing. In fact, one can argue that concepts are more primary than words and language. Then I don't think you can take results with words and use these to make statements about conceptual processing in general. In a couple of places, the authors argue that because their technique can be applied to both spatial navigation and words, it suggests that perceptual processing and conceptual processing use the same mechanisms. However, for the reason I mentioned, this does not follow. Humans certainly have a conceptual understanding of object shape as well. Can the proposed technique also capture key properties of this conceptual space?

Related to this point, not all conceptual spaces have a 2D (grid-like structure), so it is unclear why this method should work for such conceptual spaces as well.

Also, the hypothesis that perceptual and conceptual processing might be using the same mechanisms is not novel. There is a long line of research in CogSci that makes this argument, and it'd be good to mention these (for example Lakoff's work).

Other points:
- One limitation of the method is that is assumes discrete states. Can the method be extended to continuous states? It might be good to mention some ideas along these lines.
- In 4.4., in what cases would we expect addition to work? If it is only the case where the room is an exact superposition of two previous rooms, then this seems too limited.
- How about comparing against embedding from next word prediction models like GPT-3? Would these show low concept specificity?
- In 5.3, many methods can do the same vector based computation in conceptual space (even ones not based on successor representations).
- In related work, the authors say "spatial and conceptual processing can be theoretically unified into a single vector-based computational principle". This is a very strong statement not supported by the results in the paper.


**Summary Of The Paper:**

This paper presents a technique called disentangled successor information that extends successor representations (SR) in RL and shows the representations learned by the proposed method resembles grid cells in the hippocampus. The authors apply their technique to spatial navigation and language modeling and show that 1) representations for spatial navigation look like grid cells 2) representations for words resemble concept cells in the brain with each feature more or less corresponding to a single concept. They take these results to argue that perceptual and conceptual procesing in the brain could be using a similar mechanism.

In more detail, given a set of discrete states, the proposed method first forms the successor matrix S (where Sij is the discounted probability of observing state sj given state si). Then it calculates an information metric called positive successor information from this matrix. Finally, a dimensionality reduction technique based on non-negative matrix factorization is used to extract lower dimensional state representations. This dimensionality reduction technique has an additional objective term to minimize the correlation between different state vectors. This helps to create a more disentangled state representation.

The authors show that this proposed technique is related to value functions in linear RL and also to word vectors obtained by skip-gram models.


**Summary Of The Review:**

I think this is overall an interesting paper but it seems more suited for a venue with more neuroscience/cogsci focus. I don't think the results will be very interesting to the ICLR community (for that it needs much stronger empirical results on some relevant task/problem) and as it stands, the empirical results are quite limited.

---

> ### Author Response · Authors · 2022-11-18
> **Response to Reviewer 33UN (1)**
>
> We appreciate constructive and helpful comments from the reviewer. Here we provide point-to-point reply to the comments. We apologize for posting multiple replies.
>
> Comment: Overall I think this is an interesting paper but I'm not sure if ICLR is the right venue for this work. This work mainly tries to establish a connection between their approach (disentangled successor information) and perceptual and conceptual processing in the brain. This is an interesting and valuable endeavor but it's unclear how relevant this would be to the ICLR community. To me it seems like a cogsci/neuro venue or even NeurIPS would be a better choice for this work.
>
> Reply: We have seen several relating papers on theoretical framework for grid cells published in ICLR (e.g. Cueva et al., 2018; Gao et al., 2019; Whittington et al., 2022). Therefore, we thought that (at least a part of) ICLR community is interested in theoretical understanding of neural representations in the brain, and our work fits to such interest. However, we will consider other venues in case this submission would be rejected.
>
> Comment: In terms of results, there is a nice evaluation of the technique on spatial navigation and on word processing but these seem rather limited. Finding that the learned representations resemble grid cells is interesting but not a very strong evidence since many other techniques also learn representations with grid-like receptive fields.
>
> Reply: The central goal of this study is to demonstrate that we can theoretically connect computational frameworks for spatial representations (grid cells and spatial navigation) and conceptual representations (concept cells and word embedding). We agree that many other models can form grid-like representations, but it has not been shown that such model can learn intuitive and biologically plausible conceptual representations from real-world data containing complex semantic information. Furthermore, to our knowledge, concept-specific representations have not been reported in previous study on NLP models, thus that point would be novelty in our study. We emphasized this point in Section 2 (contributions and related works).
>
> Comment: Overall, I think it'd be great to have more evidence supporting the use of representations like DSI in the brain, by perhaps testing out other properties of these representations (like word similarity etc.).
>
> Reply: In the revised manuscript, we additionally mentioned correspondence to the experiment by Reber et al., 2019 (Section 3.3 and 5.3). They showed that pattern similarity of population activities of concept cells represents abstract semantic structure of objects (hierarchical structure of categories). It suggests that there are both unit-level and population-level semantic encoding in human medial temporal lobe. Our result is consistent with this experimental finding because cosine similarity of DSI vectors corresponds to word similarity. We visualized the representational structure of DSI vectors, and we could see clustering of words corresponding to semantic categories, as in Reber et al., 2019 (Appendix A.14).
> We also referred to Solomon et al., 2019, which revealed that hippocampal theta oscillation codes semantic distances between words measured in word2vec space (Discussion).
>
> Comment: One major issue with the paper for me is that it conflates conceptual processing with language processing. Language (and words) are only a part of conceptual processing. In fact, one can argue that concepts are more primary than words and language. Then I don't think you can take results with words and use these to make statements about conceptual processing in general. In a couple of places, the authors argue that because their technique can be applied to both spatial navigation and words, it suggests that perceptual processing and conceptual processing use the same mechanisms. However, for the reason I mentioned, this does not follow. Humans certainly have a conceptual understanding of object shape as well. Can the proposed technique also capture key properties of this conceptual space?
>
> Reply: As pointed out by the reviewer, we have not applied our model to general conceptual space. Therefore, we limited our scope to “linguistic concept”, and used this expression in many parts of the revised manuscript. We are planning to apply this model to other cases by extension for learning from low-level sensory inputs (as described in Discussion section), but it is left for future research.

---

> ### Author Response · Authors · 2022-11-18
> **Response to Reviewer 33UN (2)**
>
> Comment: Related to this point, not all conceptual spaces have a 2D (grid-like structure), so it is unclear why this method should work for such conceptual spaces as well.
>
> Reply: We described additional interpretations in Section 3.3, 5.3 and Appendix A.15. We speculate that constraints for the emergence of grid cells are similar to those for disentanglement of representation spaces, and the model generates disentangled word representation vectors by aligning each semantic relationship to each dimension.
>
> Comment: Also, the hypothesis that perceptual and conceptual processing might be using the same mechanisms is not novel. There is a long line of research in CogSci that makes this argument, and it'd be good to mention these (for example Lakoff's work).
>
> Reply: We referred to Lakoff’s work (conceptual metaphor) in Introduction.
>
> Comment: One limitation of the method is that is assumes discrete states. Can the method be extended to continuous states? It might be good to mention some ideas along these lines.
>
> Reply: We are planning to combine our model with unsupervised clustering model for deep learning, such as deep cluster. We expect that such extension enables learning from continuous sensory inputs (e.g. visual, auditory). We described this point in Discussion section.
>
> Comment: In 4.4., in what cases would we expect addition to work? If it is only the case where the room is an exact superposition of two previous rooms, then this seems too limited.
>
> Reply: We empirically found that composition of vectors results in composition of new metric space for the novel spatial context (Appendix A. 9). We think that this is useful for estimation of a new state transition structure in general reinforcement learning tasks (such as control of robotic arms and playing games). In the literature of reinforcement learning, it is possible to obtain appropriate Q functions for a composite task by adding Q functions of two tasks (Haarnoja et al., 2018; Makino, Accepted). We think our method can be related to or combined with such theories. We wrote this point in Section 4.4.
>
> Comment: How about comparing against embedding from next word prediction models like GPT-3? Would these show low concept specificity?
>
> Reply: We cannot analyze representations in GPT-3 because the model is not open (only interface is available). Instead, we analyzed representations in the embedding layer of pretrained BERT model, and found that conceptual specificity of BERT representations is lower than that of DSI representations (Section 5.3).
>
> Comment: In 5.3, many methods can do the same vector based computation in conceptual space (even ones not based on successor representations).
>
> Reply: The central goal of this study is not improvement of task performance but to establish theoretical connections between spatial representation model for grid cells and representation model for linguistic concepts (word embedding). On the one hand, our model can perform spatial navigation based on value-based decision making. On the other hand, our model can generate conceptual representations, by which we can perform vector-based computation like word embedding methods. We would like to demonstrate that a single model can do these two things in different computational domains. Thus, it was enough to demonstrate similar performance in section 5.3. Although other word embedding methods show similar performance, they cannot be firmly related to reinforcement learning and spatial navigation.
> Furthermore, in the revised manuscript, we included additional discussion about word representations in DSI (Appendix A.15). In that section, we argued that each word is represented by a combination of factorized conceptual units, and vector-based analogical inference corresponds to recombination of those conceptual units, in a simple case. Such property has not been observed in typical word embedding methods. Therefore, our model provides more intuitive and biologically implementable way of computation than other word embedding methods.
>
> Comment: In related work, the authors say "spatial and conceptual processing can be theoretically unified into a single vector-based computational principle". This is a very strong statement not supported by the results in the paper.
>
> Reply: We modified that sentence. In the revised manuscript, we limited our claim in linguistic concepts. Application to more general conceptual space is left for future study.

---

> ### Author Response · Authors · 2022-11-18
> **Response to Reviewer 33UN (3)**
>
> Comment: Similary for word embedding results, I didn't really find them very strong. The authors show that features in their learned representations are on average more concept specific than some alternative techniqes. Since the brain also seems to have concept specific cells, they take this as evidence for brain using a similar mechanism. However, as far as I know it is still not very clear to what extent the concept representations in the brain are like concept cells. There seems to be many cells that do not cleanly respond to a specific concept for example. Even if this was not an issue (and we knew brain used concept cells), it is still not very strong evidence because other techniques also learn representations with similar properties. In fact, one could also directly use one-hot vectors as word representations and these would be 100% concept specific.
>
> Reply: To clarify non-triviality of concept-specific representations, we described additional analyses and interpretations in the revised manuscript (Section 3.3, 5.3, Appendix A.14, A.15). We showed that word representation vectors obtained by DSI are non-sparse and distributed (Appendix A.15). It means that a word is represented by combination of concept-specific units. Furthermore, cosine similarity between vectors represents semantic similarity between words as in word embedding methods. Therefore, DSI vectors represent semantic information at both unit level and population level. This property is qualitatively consistent with concept cells because each concept cell represents a specific concept (Quiroga, 2012) and population-level pattern similarity represents abstract semantic structures (Reber et al., 2019). To the best of our knowledge, such word representations have not been found. We think it is not so trivial to obtain such representations, although we cannot exclude the possibility that we can find other approaches in the future (as mentioned in Discussion in the revised manuscript). At least, one-hot representations do not satisfy those conditions.
>
> Furthermore, concept cells were originally defined as a cell that show multimodal response to a specific concept. For example, “Luke Skywalker cell” responds to pictures, written and spoken words that indicates Luke Skywalker (Quiroga et al., 2012). Without such kind of generalization, that cell may just represent a specific sensory input, not abstract concept. Again, simple representational strategy like one-hot representations do not satisfy this requirement, whereas DSI units respond to various words related to specific concepts apparently. Therefore, we think our results of conceptual representations are not trivial as the reviewer pointed out.
>
> Even if the fraction of concept cells is not so large in the brain, we think it is still useful to theoretically investigate them. First, human entorhinal cortex is probably related to various functions (e.g. spatial navigation, conceptual learning, working memory), thus it is natural that representations for a specific function are not dominant. Next, regarding spatial representations, there are not only grid cells but also other types of representations in entorhinal cortex, e.g. head direction cells, speed cells, object vector cells and conjunctive representations. It is possible that similar diversity exists in conceptual representations, thus neurons that are apparently non-conceptual may have computational meaning. Theoretical studies like our model are useful to give interpretation to such apparently non-conceptual representations by revealing previously unknown representational forms for concepts. If we assume correspondence between grid cells and concept cells, we may be able to extend our approach to predict counterparts of head direction cells and speed cells for language, for example. We think such prediction is useful to expand our understanding of conceptual representations in the brain, because exhaustive experimental investigation like animal experiments is not possible in human experiments. We think our model can be a first step towards such prediction.

---

### Official Review · Reviewer_Eei5 · 2022-10-24

**Confidence:** 4
**Clarity, Quality, Novelty And Reproducibility:** The clarify, quality and rigor needs …
**Correctness:** 2
**Technical Novelty And Significance:** 3
**Empirical Novelty And Significance:** 2
**Recommendation:** 3

**Strength And Weaknesses:**

Strength:
* The paper studies both grid cells and word embedding problems.
* The attempt to link grid cells and word embedding is ambitious.
* Using successor information to study the grid cells is an interesting and novel idea.



Main concerns:
* Why decorrelative NMF is a good objective function for the grid cell system is unclear. Eq. (6) needs better motivations.

* It is unclear what determines the structure of the emerging representation. Does the word embed similarly lead to a grid representation? If not, why?

* The improvement of this model (in the case of word embedding ) compared to prior work is in fact very subtle (e.g., Table 2).

* How is the present work different from prior work, e.g., Stachenfeld et al 2017, and Dordek et al 2016 is unclear. This paper performs matrix factorization, while previous work did PCA. Are they really different? This needs to be better explained.

* The similarity between Eq 13 and Eq 15 seems to be superficial. If they are indeed equivalent, the implication seems to be that it should be possible to use an objective function based on Eq. 13 to derive the grid cells.  But this is not discussed or shown.

* Interpreting an information-theoretical quantity as the neural activity seems to be risky. (I am actually not sure if it makes any sense.) This crucial assumption needs to be better justified.


Other comments:
In Dordek et al., 2016; Sorscher et al., 2019, in addition to the non-negativity, the inhibition surround is also critical for the grid firing fields to emerge. This point should be discussed in more detail.
Several relevant papers on understanding grid cells with machine learning approaches are missing, e.g.,  Cueva & Wei, ICLR, 2018;  Gan, Xie, Zhu, Wu, ICLR, 2019.
In Section 4.2, this explanation is not the first that attempts to theoretically explain the grid responses and the scale ratio. Prior work on this should be discussed.

**Summary Of The Paper:**

The authors develop a “disentangled successor information” framework for understanding the neural representation for physical and conceptual spaces. The paper then applied this framework to study two problems: grid cells and word embedding.
Overall, this paper contains some interesting and intriguing ideas. But the technical content is not entirely rigorous.




**Summary Of The Review:**

Overall, I enjoyed reading this paper, but there are many question marks upon reading it. I’d be more comfortable endorsing this paper if the work is done in a more rigorous way, and the model assumptions are better justified. Right now, there is uncertainty about the novelty of this paper makes and whether the results are indeed solid.


%%%%%%
I would like to thank the authors for their detailed responses to my comments. Although the responses and the revision clarified some issues, unfortunately, they did not lead to major improvements in the paper in my opinion. The paper has nice potential to be further improved, but at the moment the contribution still seems rather incremental, and the work would benefit by making the arguments more rigorous.

---

> ### Author Response · Authors · 2022-11-18
> **Response to Reviewer Eei5**
>
> We appreciate constructive and helpful comments from the reviewer. Here we provide point-to-point reply to the comments. We apologize for posting multiple replies.
>
> Comment: Why decorrelative NMF is a good objective function for the grid cell system is unclear. Eq. (6) needs better motivations.
>
> Reply: We explained this point in Section 3.3 of the revised manuscript. We chose those constraints because they are related to the emergence of grid-like representations and disentangled representations. Please check revised manuscript for the detail.
>
> Comment: It is unclear what determines the structure of the emerging representation. Does the word embed similarly lead to a grid representation? If not, why?
>
> Word representations are not grid-like in our study (at least we have not found grids). Our model can be applied to arbitrary inputs, and generates grid-like representations if the input is sequences in the 2-D structure, and generates concept-specific representations if the input is word sequences. Those two are different, which means that our model changes the representational form depending on the structure of inputs. From this result, we claimed that there can be shared learning principle behind apparently different representations for 2-D spaces and concepts. It is known that grid patterns are disordered even in 3-D space (Ginosaur et al., Nature, 2021; Grieves et al., Nature Neuroscience, 2021), thus we speculate that grid-like representations are quite specific in flat 2-D spaces. Outside of that condition, same mechanism can generate different representational form.
>
> Although the mathematical mechanism behind grid-like representations in 2-D space have been analyzed in previous studies (Sorcher et al., 2019; Gao et al., 2019), we have not completely elucidated the mechanism behind concept-specific representations. However, we described additional interpretations in Section 3.3, 5.3 and Appendix A.15. We speculate that constraints for the emergence of grid cells are similar to those for disentanglement of representation spaces, and the model generates disentangled word representation vectors by aligning each semantic relationship to each dimension.
>
> Comment: The improvement of this model (in the case of word embedding ) compared to prior work is in fact very subtle (e.g., Table 2).
>
> Reply: The central goal of this study is not improvement of task performance but to establish theoretical connections between spatial representation model for grid cells and representation model for linguistic concepts (word embedding). On the one hand, our model can perform spatial navigation based on value-based decision making. On the other hand, our model can generate conceptual representations, by which we can perform vector-based computation like word embedding methods. We would like to demonstrate that a single model can do these two things in apparently different computational domains. Therefore, it was enough to demonstrate similar performance in section 5.3. Although other word embedding methods show similar performance, they cannot be firmly related to reinforcement learning and spatial navigation.
>
> Furthermore, in the revised manuscript, we included additional discussion about word representations in DSI (Appendix A.15). In that section, we argued that each word is represented by a combination of factorized conceptual units, and vector-based analogical inference corresponds to recombination of those conceptual units, in a simple case. Such property has not been observed in typical word embedding methods. Therefore, our model provides more intuitive and biologically implementable way of computation than other word embedding methods.
>
> Comment: How is the present work different from prior work, e.g., Stachenfeld et al 2017, and Dordek et al 2016 is unclear. This paper performs matrix factorization, while previous work did PCA. Are they really different? This needs to be better explained.
>
> Reply: As we wrote in section 2 of the revised manuscript, we did not aim to build completely new model of grid cells. The main contribution of our study is to demonstrate that we can extend the theoretical framework for grid cells to biologically plausible conceptual representations (concept cells) despite they are apparently different representational form. Therefore, we basically followed the existing literature of grid cell models (we based our model on SR, and used constraints related to the emergence of grid cells) although we performed some mathematical extensions to connect it to learning of linguistic concepts. Specifically, we extend SR to SI, which added connection to word embedding (PMI). Furthermore, we used decorrelative NMF, which was necessary to obtain both grid cells and concept-specific units. Although each component in decorrelative NMF follow existing literature of grid cells, we found the specific mathematical form by which we can generate both spatial and conceptual representations in EC.

---

> ### Author Response · Authors · 2022-11-18
> **Response to Reviewer Eei5 (2)**
>
> Comment: The similarity between Eq 13 and Eq 15 seems to be superficial. If they are indeed equivalent, the implication seems to be that it should be possible to use an objective function based on Eq. 13 to derive the grid cells. But this is not discussed or shown.
>
> Reply: Coincidence probability in Eq. 13 (Eq. 8 in the revised manuscript) can be arbitrarily defined. Word embedding methods often use symmetric rectangular window for its simplicity. However, if we evaluate coincidence by asymmetric exponential kernel, we obtain the form of SI. Therefore, we may interpret SI as a special case of PMI by regarding states as words. We included detailed discussion of the relationship between PMI and SI in Appendix A.4.
>
> Comment: Interpreting an information-theoretical quantity as the neural activity seems to be risky. (I am actually not sure if it makes any sense.) This crucial assumption needs to be better justified.
>
> Reply: We speculate that a neural network model can learn DSI representations. We related DSI representations to PMI, and dimension reduction of PMI gives optimum of skip-gram neural network (Levy and Goldberg, 2014). Therefore, we can expect that DSI representations is obtained by combination of skip-gram neural network with SR, decorrelation and non-negativity constraints. Such neural network is reasonable because hippocampal representations can be regarded as SR (Stachenfeld et al., 2017), and decorrelation and non-negativity are biologically plausible constraints. We described this possibility in Discussion section.
>
> Furthermore, previous animal experiments have revealed the existence of neural activities corresponding to Shannon information (Nakamura, J Neurophysiol, 2006) and log-likelihood ratio (Kira et al., Neuron, 2015). Therefore, we think it is not so unrealistic to find neural activity corresponding to probabilistic or information-theoretical quantity.
>
> Comment: Other comments: In Dordek et al., 2016; Sorscher et al., 2019, in addition to the non-negativity, the inhibition surround is also critical for the grid firing fields to emerge. This point should be discussed in more detail. Several relevant papers on understanding grid cells with machine learning approaches are missing, e.g., Cueva & Wei, ICLR, 2018; Gan, Xie, Zhu, Wu, ICLR, 2019. In Section 4.2, this explanation is not the first that attempts to theoretically explain the grid responses and the scale ratio. Prior work on this should be discussed.
>
> Reply: We included suggested references, and additionally referred to previous results on orthogonality and scale ratios in Section 2 and 3.3.

---

### Official Review · Reviewer_K3nY · 2022-10-25

**Confidence:** 4
**Correctness:** 3
**Technical Novelty And Significance:** 3
**Empirical Novelty And Significance:** 3
**Recommendation:** 6

**Clarity, Quality, Novelty And Reproducibility:**

(1) The paper is very clearly written.

(2) The theoretical foundation of the paper is solid.

(3) The proposed method is new, but is not entirely original, given the past work on successor representation for place cells and past work on word embedding. Matrix factorization is an element in both areas. Word embedding in NLP has gone far beyond matrix factorization. Dimension reduction may not explain grid cells either given the large number of grid cells in EC.

**Strength And Weaknesses:**

Strengths:

(1) The theoretical foundation of the paper is solid, especially relationship with linear reinforcement learning.

(2) The empirical results on grid cells are strong.

(3) The experiments on word embedding are solid.

Weaknesses:

(1) Non-negative matrix factorization is widely used, such as in the recommender system, to factorize the value function or the affinity matrix. The successor representation has also be used to model place cells, which are connected to grid cells via matrix factorization or eigen decomposition. Thus this paper is not entirely novel, although the specific form of decorrelative NMF is new.

(2) Given the large number of grid cells in EC, dimension reduction may not be the most convincing point of view for embedding. It is more about population coding, where generalization does not need to rely on dimension reduction. Other forms of regularizations may also work.

(3) Can your method on grid cells explain path integration?

(4) The sequence of words in NLP is not a trajectory. The contexualized embedding such as BERT seems more reasonable.

**Summary Of The Paper:**

This paper defines positive successor information in the framework of reinforcement learning, and proposes positive decorrelative nonnegative matrix factorization for dimension reduction of successor information. The paper applies the proposed method to spatial navigation and word embedding, and obtained meaningful results. The paper also analyzes theoretical relationship to linear reinforcement learning.

**Summary Of The Review:**

The paper makes a solid and useful contribution, but it is not extremely novel.

---

> ### Author Response · Authors · 2022-11-18
> **Response to Reviewer K3nY**
>
> We appreciate constructive and helpful comments from the reviewer. Here we provide point-to-point reply to the comments.
>
> Comment: (1) Non-negative matrix factorization is widely used, such as in the recommender system, to factorize the value function or the affinity matrix. The successor representation has also be used to model place cells, which are connected to grid cells via matrix factorization or eigen decomposition. Thus this paper is not entirely novel, although the specific form of decorrelative NMF is new.
>
> Reply: As we wrote in section 2 of the revised manuscript, we did not aim to improve task performance or build completely new framework for grid cells. The main contribution of our study is to demonstrate that we can extend the theoretical framework for grid cells to biologically plausible conceptual representations (concept cells) despite they are apparently different representational form. Therefore, we basically followed the existing literature of grid cell models (we based our model on SR, and used constraints related to the emergence of grid cells) although we performed some mathematical extensions to connect the model to learning of linguistic concepts. To the best of our knowledge, our study is the first one that proposed the theoretical relationship between grid cells and concept cells (and also, spatial navigation and word embedding). Previously, grid cell model have not been applied to complex conceptual spaces in real world, and word embedding methods did not exhibit interpretable unit-level conceptual specificity.
>
> Comment: (2) Given the large number of grid cells in EC, dimension reduction may not be the most convincing point of view for embedding. It is more about population coding, where generalization does not need to rely on dimension reduction. Other forms of regularizations may also work.
>
> Reply: We think dimension reduction should happen even if the number of grid cells is larger than the number of place cells. It is because hippocampal place cells rapidly change their representations and remap across contexts, in contrast that grid cells (and other spatial representations in EC) are basically persistent across contexts. Therefore, dimensionality of representations in the long term should be larger in hippocampus than EC. Although we admit that dimension reduction may not be the most important (we have not proven that), we think it is natural to assume dimension reduction in our model.
>
> Comment: (3) Can your method on grid cells explain path integration?
>
> Reply: We performed path integration based on DSI representation vectors, using movement-conditional recurrent weights (Gao et al., 2019; Oh et al., 2015). Results are shown in Appendix A.7.
>
> Comment: (4) The sequence of words in NLP is not a trajectory. The contexualized embedding such as BERT seems more reasonable.
>
> Reply: As the reviewer pointed out, our current model is context-independent embedding, and it should be extended to process context-dependence. We are planning to combine our model with other methods such as clone-structured cognitive graph (George et al., 2021) to enable learning of context-dependent latent states. We wrote this point in Discussion section.
>
> We did not use transformer-based model because it is hard to interpret representation vectors theoretically. We discussed mathematical relationship between successor information, value functions and word embedding. That was possible because mathematical properties of word embedding methods have been firmly elucidated, but such interpretation is currently difficult for transformer-based models. We wrote this point in Section 2 (Contributions and related study).
>
> We also showed that representations in BERT model do not show high conceptual specificity (Section 5.2). Therefore, BERT (at least in its current form) do not generate intuitive conceptual representations like our model. However, we do not exclude possibility that we can obtain similar results using extended transformer-based models. Even in that case, our study would be useful to consider how to build such model.

---

### Author Response · Authors · 2022-11-18
**Summary of revisions and general response to reviewers**

We thank all reviewers for their constructive comments. Those comments were helpful for us to improve our paper. We summarize our revisions below. We hope that these revisions would resolve concerns raised by reviewers.

* Several reviewers raised concern about novelty and performance of our model. For clarification, we significantly revised the section 2 (contributions and related study). We explicitly described scope and contribution of our study and added several references.
As described there, we summarize contributions of this work as follows. (1) We extended SR to successor information (SI), by which we theoretically connected reinforcement learning and word embedding, thus spatial navigation and conceptual inference. (2) We found that dimension reduction with constraints for grid-like representations (decorrelative NMF) generates disentangled word vectors with concept-specific units, which has not been found previously. (3) Combining these results, we demonstrated that the computational principle for grid cells can be extended to represent and compute linguistic concepts (one of complex conceptual spaces in the real world) in an intuitive and biologically plausible manner, which has not been shown in previous studies.
We would like to emphasize that the central goal of this study is not improvement of task performance or building completely new model for grid cells, but to establish connections between grid-like spatial representations (that is consistent with previous theories and experiments) and representations of linguistic concepts, and also, spatial navigation and inference of linguistic concepts.

* Several reviewers asked about the relationship to transformer-based models such as BERT and GPT. Regarding this point, we made several revisions: (1) We analyzed conceptual specificity of representations in the embedding layer of pretrained BERT model, and confirmed that BERT representations are less concept-specific than DSI. (Section 5.2) (2) We wrote that the theoretical interpretation of our model was possible because mathematical properties of word embedding methods have been firmly elucidated, but such interpretation is currently difficult for transformer-based models. (Section 2)

* Several reviewers asked about the mechanism of the emergence of concept-specific representations. To clarify that, we described additional analyses and interpretations in Section 3.3, 5.3 and Appendix A.15. We described that word vectors obtained by DSI represent semantic relationship in both unit level and population level. We speculate that such representations emerged because constraints for the emergence of grid cells are similar to those for disentanglement of word representation spaces. The model probably generates disentangled word representation vectors by aligning each semantic relationship to each dimension.

* We showed that we can perform path integration based on DSI vectors (Section 4.3 and Appendix A.7), using the strategy proposed in the previous studies (Gao et al., 2019; Oh et al., 2015)

* We described the interpretation of constraints in decorrelative NMF (Section 3.3). We chose those constraints because they are related to the emergence of grid-like representations and disentangled representations.

* We mentioned about neural network implementation of our model (Discussion).

* As a reviewer pointed out, we have not applied our model to general conceptual spaces. Therefore, we used the expression “linguistic concepts” instead of “conceptual space” in many parts of the revised manuscript (including the title and abstract). By this expression, we clarified that we applied our model to language, which is one of many complex conceptual spaces in the real world. We discussed about extension to generalize our model in Discussion section.

* We additionally mentioned about consistency with experiments on human brain. (1) We showed clustering of words corresponding to semantic categories in the representational structure of DSI vectors, as in Reber et al., 2019 (Appendix A.14). (2) We referred to Solomon et al., 2019, which revealed that hippocampal theta oscillation codes semantic distances between words measured in word2vec space (Discussion).

* We showed that composition of vectors results in composition of new metric space for the novel spatial context (Appendix A. 9). We mentioned about possibility of application to general reinforcement learning tasks, and the relationship with composable reinforcement learning (Haarnoja et al., 2018; Makino, Accepted) (Section 4.4).

* We wrote that w(s) also became grid-like representations (Section 4.2)

---

### Decision · Program_Chairs · 2023-01-20

**Decision:**

Reject

**Justification For Why Not Higher Score:**

This paper reiterates previously reported findings.   There is an attempt to draw some connections between those findings, but the argument is not well laid out and so falls flat.  I don't think this paper should be accepted, even though it did get one high rating.

**Justification For Why Not Lower Score:**

n/a

**Metareview: Summary, Strengths And Weaknesses:**

This paper attempts to connect the worlds of spatial navigation to language via their new method called disentangled successor information.  They show that their model creates representations that can be mapped to concepts (places, semantic concepts) for both application areas.  The reviewers questioned the novelty of this approach, as this paper mostly reiterates finding that have been previously reported in the two domains.  I appreciate the effort the authors put in to find a connection between the two conceptual spaces, but agree with the reviewers that this connection is in need of clarification.  I recognize that this paper did receive one very high score, but the reviewer may not have been aware of the work in NLP that overlaps heavily with the findings of this paper.
Though the connection between navigation and conceptual spaces is interesting and novel, it needs further development to be more impactful.


Side note:  the combination of interpretable dimensions to describe a concept has been seen before (e.g. Murphy et all 2012, tables 3 and 4 https://www.cs.cmu.edu/~bmurphy/NNSE/nnse_coling12.pdf)